# Measurements, Analysis, Classification, and Detection of Gunshot and Gunshot-like Sounds

**DOI:** 10.3390/s22239170

**Published:** 2022-11-25

**Authors:** Rajesh Baliram Singh, Hanqi Zhuang

**Affiliations:** Department of Electrical Engineering and Computer Science, Florida Atlantic University, Boca Raton, FL 33431, USA

**Keywords:** gunshot, gunshot-like, random forest, uniform manifold and projection, SHapley Additive exPlanations, Mel-frequency cepstral coefficients

## Abstract

Gun violence has been on the rise in recent years. To help curb the downward spiral of this negative influence in communities, machine learning strategies on gunshot detection can be developed and deployed. After outlining the procedure by which a typical type of gunshot-like sounds were measured, this paper focuses on the analysis of feature importance pertaining to gunshot and gunshot-like sounds. The random forest mean decrease in impurity and the SHapley Additive exPlanations feature importance analysis were employed for this task. From the feature importance analysis, feature reduction was then carried out. Via the Mel-frequency cepstral coefficients feature extraction process on 1-sec audio clips, these extracted features were then reduced to a more manageable quantity using the above-mentioned feature reduction processes. These reduced features were sent to a random forest classifier. The SHapley Additive exPlanations feature importance output was compared to that of the mean decrease in impurity feature importance. The results show what Mel-frequency cepstral coefficients features are important in discriminating gunshot sounds and various gunshot-like sounds. Together with the feature importance/reduction processes, the recent uniform manifold approximation and projection method was used to compare the closeness of various gunshot-like sounds to gunshot sounds in the feature space. Finally, the approach presented in this paper provides people with a viable means to make gunshot sounds more discernible from other sounds.

## 1. Introduction

Considering the recent uptick in senseless shootings in otherwise quiet and relatively safe environments, there is a need, now more than ever, to deter these incidents. Although the first barrier to diminishing gun violence involves the push for implementing gun control laws, artificial intelligence (AI) can play a significant role in helping deter individuals who might slip through the first barrier. The installation of sensors can assist in the proper surveillance of surroundings tied to public safety, which is the first step toward AI-driven surveillance. With the increase in popularity of machine learning (ML) processes, systems are being developed and optimized to assist personnel in highly dangerous situations. Together with saving innocent lives, helping capture the responsible criminals is part of the AI algorithm that can be hosted in acoustic gunshot detection systems (AGDSs).

Researchers have also been very active in seeking effective tools to combat gun violence. Population health technology (PHT), which is defined as the application of emerging technology to improve the health of populations [1], discusses the use of sensors in detecting acoustics related to gunshots. It is a common practice to use video for criminal surveillance and monitoring; however, this practice has its limitations. For example, the strength of video-only surveillance is inhibited by the field of view (FOV) being occluded or by a lack of proper lighting [2,3,4,5]. In these instances, videos fail to reliably detect and account for activities involving gunshots and gunshot-related crimes. The inclusion of audio analysis can complement video-based systems.

With networked sensors mounted on high structures, away from the general population’s reach to limit the occurrence of tampering, the position of a gunshot acoustic signature can be determined to within a few feet by triangulation. Mere seconds pass from the time of the gunshot incident to the activation of the alert system. Thus, acoustic systems could potentially increase the deterrent effect of police and therefore reduce the occurrence of gun-related crimes [6,7]. Another major contributing factor to the implementation of audio-based sensing is its low computational cost compared to video-based sensing. Audio analysis, in this context, can be much more easily geared toward abnormal event detection, source localization, and tracking.

The aim of this research project is to analytically compare gunshot and gunshot-like sounds in order to facilitate the inclusion of AI-driven gunshot detection technology (GDT) into AGDSs. The main objective of this study is to first characterize the closeness of the gunshot and gunshot-like sounds and then use the result to help isolate gunshot sounds from other, innocuous sound events. Several articles in the literature are geared toward the mechanics [8,9] and detection of gunshot sounds [10,11,12,13,14]. Our work here is to analyze different acoustic signatures of gunshot and gunshot-like sounds, which in turn guides the design of an effective AGDS. Although the presented results do not resolve the issue of totally isolating a gunshot sound from other, gunshot-like sounds, it does bring to light some of the important features extracted from a Mel-frequency Cepstrum Coefficient (MFCC) analysis that are important for random forest (RF) classification. This result, in turn, can help improve the performance of an AGDS as reflected by the receiver operating characteristic (ROC) and accuracy results. We present here some insights into the important features and their effect on the model’s output.

The paper is organized as follows. The motivation, together with the past and present work being carried out in efficient GDT, is presented in Section 1. Data procurement and preprocessing are discussed in Section 2. A brief review of the MFCC feature extraction algorithm together with samples of power spectral density (PSD) visuals are also included in Section 2. Section 3 presents some of the tools used in this analysis: SHapley Additive exPlanations (SHAP) and the mean decrease in impurity (MDI). Section 4 takes a look at the clustering of gunshot-like sounds in relation to the gunshot sounds via the use of uniform manifold and projection (UMAP). Section 5 discusses the classification of the audio files via the use of an RF classifier. This section also applies instruments such as ROC curves and the confusion matrix (CM) to assess the performance of the two feature importance processes for separating gunshot sounds from gunshot-like sounds. Detection of a gunshot sound in relation to gunshot-like sounds is analyzed in Section 6. Finally, Section 7 provides a summary and concluding remarks on the work presented in this paper.

## 2. Data Measurement and Preparation

As discussed in Ref. [15], to develop a robust predictive model, audio samples must be of high quality. For this purpose, we have collected gunshot-like sounds in different environments [16]. Additionally, in an effort to maintain a standardization of the data used to generate the model, scaling was carried out in the cepstral domain (see Section 5.2 for details). In addition to gathering highly representative audio files to complement the files from the open source arena [17,18,19], audio files generated from reliable sources such as Refs. [20,21,22] were utilized in this study. These complementary resources provide curated audio clips captured with sensing equipment in various environments. The audio clips are used mostly in the movie, TV, and gaming industries. Companies that are interested in machine learning have also focused on these latter resources.

In this section, we first summarize the process and results of collecting gunshot-like sounds in this research. We then organize gunshot-like sounds into audio classes together with gunshot sounds. An effective feature extraction procedure, MFCC, is outlined next with its application to the analysis of gunshot-link sounds.

### 2.1. Gunshot-like Sound Measurement

Together with procuring the gunshot and gunshot-like sounds from both the open and paid resources listed above, the authors generated their own database for the plastic bag pop (class 8 in Table 1) gunshot-like sound.

The plastic bag pop sounds were recorded in likely environments where gunshots can be fired, some of which are listed here (see Figure 1 below): (a) inside a building along a corridor, (b) inside a personal dwelling, (c) outdoors between two buildings, (d) outdoors on the side of a building and, (e) outdoors in an open field. Together with these likely environments, a controlled set of data was taken in an anechoic chamber (see Figure 1f below).

It is noted in the Figure 1 above, that together with the various environments, various microphones were used in the data collection process. The various microphones and their associated frequency response enhanced the audio variety of the collected data.

Together with the two variables listed above, that is, the environment and the microphones, various sizes of plastic bags and distances from the microphones were also included during the collection process. The variety of audio files collected in this procedure lends to a very robust dataset. In the experiments, the Tascam DR-05X, Zoom H4nPro, Brüel & Kjær, Blue Yeti USB, JLab TALK GO, iPad mini, and a Samsung S9 were the recording devices used. The Samsung S9 was the only device that automatically adjusted its recording level, while the others had to be manually adjusted. These different recording devices, with its various sensitivity in frequency range, allowed for the capture of plastic bag pop sounds with various audio responses and audio levels. Much more granular details of the data collection process can be found in Ref. [16].

### 2.2. Audio Classes

As previously mentioned, audio samples of gunshot-like sounds were procured. Table 1 below shows the order of the audio classes together with a partial list of the groups of audio clips in each class used in the analysis.

Our initial analysis consisted of 250 audio files each for the following classes: (0) carbackfire, (1) cardoorslam, (2) clapping, (3) doorknock_slam, (4) fireworks, (5) glassbreak_bulb-burst, (6) gunshots, (7) jackhammer, (8) plastic_pop, and (9) thunderstorm. The classes chosen here were procured as per the research done by the authors in Refs. [6,23,24,25,26,27].

For the majority of the gunshot clips in the database, the actual gunshot or gunshot-like sound lasted for approximately 0.3 s. With this in mind, the authors decided to standardize, to 1 s, the duration of all the audio clips to be fed to the feature extraction process discussed next.

### 2.3. Feature Extraction

There is a wide variety of feature extraction methods to choose from. The feature extraction method chosen for this analysis is the MFCC. Although the MFCC was initially implemented for speech recognition [28], it has found its way to the realm of sound classification [29,30,31].

Figure 2 below shows the normalized Mel-filter bank that is typically implemented in a 40 MFCC feature extraction process of a 96 k sample rate audio file (note that although a sample rate of 48 kHz is more than sufficient for the analysis that follows, the 96 k sample rate example is used just as an illustration since some of the audio files from the secondary databases were sampled at 96 kHz). In a conventional application, as done in librosa, the end frequency of the filter bank is set to sample rate/2. From the minimum to the maximum filter bank frequencies, 40 triangular filters are produced and resized according to the Mel-scale.

An email (B. McFee, personal communication, 22 July 2022) conveyed information that adding the delta features accounts for local, short-term temporal patterns. Having only the MFCC feature extraction will cause the analysis to become sensitive to exact timing alignment, which is an area that should be avoided, especially when the dataset is limited.

Figure 3 shows the block diagram for MFCC feature extraction together with its delta coefficient (Δ—differentiation of MFCCs, called the velocity features) and delta-delta coefficient (ΔΔ—double differentiation of MFCCs, called the acceleration features). Stated another way, the Δ and ΔΔ are approximations of the first and second temporal derivatives of MFCCs.

Note that one of the major pitfalls of using the Δ features is that differentiators tend to amplify noise. This effect causes the output to be noisier than the original signal. Differentiation applied twice causes the features to be even more unstable.

In our analysis, we extracted 40 features for the MFCC, Δ and ΔΔ, respectively. We then took the mean and standard deviation of each extracted feature and stacked the vector horizontally per sample. This process obtained 240 features in total (see Figure 4a–c later in the paper). The act of using the mean and standard deviation features was done to increase the feature set for feature importance/feature reduction analysis.

As we take a closer look at the extracted MFCC features, we will see, later on in this paper, (refer to Section 3 below) some of the important features sorted as per the relevant feature importance algorithm.

Table 2 below shows the feature names together with their respective feature meanings. For example, feature 0 = MFCC_MEAN_FLTR0, which means MFCC mean coefficients from Mel band 0 (from 0 Hz–160.97 Hz). Another example: feature 200 = DELTA2_STDDEV_FLTR0, which means delta-delta standard deviation coefficients from Mel band 0.

It is worth noting here that Mel bands will change depending on the sample rate of the input audio file and also the number of MFCCs chosen. Table 3 below lists the Mel triangular frequency bands used for 96k sample rate audio files. Additionally used for the generation of this frequency table is the number of Mel coefficients. As mentioned earlier, 40 Mel coefficients/filters were implemented in our analysis.

When one is using MFCC as the feature extraction process, good resolution at low frequencies is attained, whereas at higher frequencies, broad ranges get lumped into one band. This is because the Mel-scaled bank is designed to mimic the human auditory system. Human perception of pitches, which can be approximately described as logarithmic, translates into better deciphering of low frequencies as compared to high frequencies. The MFCC feature vector represents the spectral envelope of a single frame in the sense that two signals with a similar spectral envelope will have a similar sequence of MFCCs.

As can be observed from the zoomed in spectrogram plots of the plastic bag pop and the gunshot sound in Figure 5 below using Adobe Audition, much of the energy content of the gunshot sound is concentrated in the lower end of the frequency spectrum.

We will show later in Section 3 how this observation of the low frequency content concentration leads to MFCC feature importances, which are also concentrated in the low end of the frequency band.

### 2.4. Power Spectral Density

Figure 6 below shows a sample from each of the 10 classes. Each figure (Figure 6a–j) shows the spectrogram, together with the amplitude over time (to the bottom of the spectrogram) and the power spectral density (PSD) (to the left of the spectrogram). In this view, noting that all the frequency content is in the power spectrum, the spectrogram tells us where in time those frequencies occurred. In addition, we observe that the power spectrum is the cumulative average of the spectrogram, averaged over time. We can physically see where in time the majority of the frequency content is concentrated.

Using Figure 6g (the gunshot sound) as an example, we can conclude that the plastic_pop sound (Figure 6i), looks very similar from the PSD perspective, that is, a lot of energy is concentrated in the 0–10 kHz region, and then quickly decays to about half its original spectral density.

Figure 7 provides a physical (waterfall plot) view of the same samples given in Figure 6 above, for each of the various classes. Each plot shows the power spectral density (PSD) over time and frequency. In Figure 7g, there are two gunshot sounds in quick succession at the beginning of the audio clip. If we take just a single shot and compare that power spectral density to the plastic-pop sound (Figure 7i), we see some similarities, confirming our observation from Figure 6g,i above.

## 3. Analysis Tools

Coupled with the low-level feature engineering process (i.e., simply generating the mean and standard deviation of the MFCCs and their derivatives), feature importance is also analyzed. Based on the feature importance analysis, we then employ feature selection.

As a general rule of thumb, if one has more features than samples, one runs the risk that the observations will be harder to cluster. According to the Hughes phenomenon [32], as the number of features increases, the classifier’s performance also increases, until the optimal number of features is attained. Adding more features beyond the size of the training set will then degrade the classifier’s performance. To overcome this curse of dimensionality, we apply two popular feature selection/reduction techniques—Mean Decrease in Impurity and Shapley Additive Explanations—to the analysis of gunshot-like sounds.

### 3.1. Mean Decrease in Impurity

RF is an ensemble-trees model used mostly for classification. Ensemble methods combine several decision trees to produce a better predictive performance than that of a single decision tree.

The gini impurity measure, one of the methods used in decision tree algorithms, determines the optimal split from a root node and its subsequent splits. The gini impurity of a dataset is a number between 0 to 0.5. It denotes the probability of misclassifying an observation. The lower the gini impurity, the better the split, and the lower the likelihood for misclassification. When all cases in the node fall into a single target category, a value of 0 is attained. The mean decrease in impurity (MDI) is the added weighted impurity decrease for all nodes and the average over all trees.

Figure 4a–c above shows the mean and standard deviation of the MFCC features extracted via the librosa library [33] and transformed via the sklearn RF MDI. The mean of the delta and delta-delta features does not add much to the overall feature importance, although the standard deviation does show some impact in the early part of the feature extraction process. As a refresher, designations of feat_”X” (or feature “X”) refer to the “X” Mel-triangular filter or band (see Table 3 above) that is processed during the MFCC calculations.

Figure 4d above shows the 20 most important features sorted by the RF feature importance based on MDI. Figure 4d above displays no features beyond the frequency band of 3807.86 to 4844.28 Hz (see Table 2 and Table 3 above) are present.

The plot in Figure 8 below shows the distribution of the data using the box and whisker approach for the 20 most important MDI features. The data in Figure 8 is displayed in quartiles, which also includes the outliers. Features 2 and 3 show approximately the same dispersion of data as well as the interquartile range (length of the box). Additionally, the overall spread of features 2 and 1 is a bit more compared to the other features (length of the whiskers). The model’s prediction towards the uncertainty region is guided by the outliers. An increase in outliers results in greater uncertainty of the model. Finally, a positive or right skew of the box plot indicates that higher values of the feature occur more often.

Considering the three most important features (features 2, 3, and 40 as determined by the RF MDI feature importance) for Figure 9 below, in both the 2D and 3D space, the features for gunshot, jackhammer, doorknock_slam, and glassbreak_bulb-burst seem to be fairly dispersed. The remainder of the classes appears to be fairly defined even down to a 2D space. The gunshot sounds appear to be closely intertwined with the plastic_pop sounds. Taking the centroids of each class (see Figure 9a above) and comparing the relative distances from the gunshot sounds, we arrive at the results shown in Table 4 below. In Table 4 below, it can be observed that the plastic bag pop sound is closest to that of the gunshot sound [16] (note that “feat2 ” and “feat3” is the *x*, *y* coordinate respectively of the centroids of each class (see Figure 9a above)). Additionally shown in Table 4 below, is a comparative analysis, indicating which sound is closest to the gunshot sound in descending order, namely: plastic bag pop, door knock, door slam, and car backfire. Additionally, researches also stated that the door knock, door slam, and car backfire was close to the actual gunshot sound [23,24,25,34,35,36,37,38].

### 3.2. SHapley Additive exPlanations

SHapley Additive exPlanations (SHAP) is a methodology that can be used to interpret a model. Named after Lloyd S. Shapley, the SHAP concept was originally developed to estimate the importance of an individual player on a collaborative team. This concept was geared toward distributing the total gain or payoff among players, depending on the relative importance of their contributions to the outcome of a game [39].

Application of SHAP in ML includes the “total gain” or “payoff” as the model prediction (f(x)) for a single instance of the dataset, and the “players” as the features of the instance that collaborate to receive a gain (predicted value). The SHAP values are the averaged marginal contribution of a feature value across all possible coalitions.

Although beyond the scope of this paper, it is worth noting that in addition to giving us the ability to extract important features, SHAP falls into the category of interpretable machine learning (IML). IML aims to build models that can be understood by humans.

SHAP feature importance is model-agnostic compared to model-specific, as in the MDI feature importance carried out above. Model-agnostic feature importance can, in principle, be used for any model. The algorithm is treated as a black box that can be swapped out for any model. Methods involving model agnostic evaluations provide flexibility when it comes to model selection. Different models can employ the same evaluation framework. In this manner, a comparison of many models can be carried out using the same metrics. Maintaining a consistent framework allows for a much more robust comparison between models.

Another important distinction between the SHAP process and the MDI is the availability of local and global feature importance. Local feature importance focuses on the contribution of features for a specific prediction, whereas global feature importances take all the predictions into account.

Figure 10 below shows the sorted feature importance as per the SHAP calculations. Each bar shows the contribution each class has to the model’s output. Focusing on class 6 (colored light blue for emphasis), we see the contribution that the gunshot sound makes to each of the 20 most important features shown in the figure. Note in Figure 10, that for feature 80 (or the mean of the first delta coefficient - DELTA_MEAN_FLTR0), the gunshot sound is the most dominant contributor.

Table 5 compares the 20 most important features as calculated by the RF MDI and SHAP processes. The red colored cells show the features that are different between the both processes. Although the MDI and SHAP show different orders of the features’ importance, the first 11 features, are common to both, but have a different order. We will see later on in Figure 11 in Section 5.3, that by using only these 20 features, we can achieve an accuracy of approximately 92%.

## 4. Gunshot Proximity Analysis

Before delving into the classification phase of this project, let us investigate the proximity of the gunshot-like sounds to the gunshot sounds. To assist in our analysis, we employ the uniform manifold approximation and projection (UMAP) method.

### UMAP

UMAP is a novel manifold learning technique for dimension reduction. It is constructed from a theoretical framework based on Riemannian geometry and algebraic topology [40]. Although it has a rigorous mathematical foundation, it is easy to use via the scikit-learn compatible API.

In its simplest sense, the UMAP algorithm consists of two steps: (1) construction of a graph in high dimensions and (2) optimization to find the most similar graph in lower dimensions.

UMAP is among the fastest manifold learning implementations available and is significantly faster than most t-distributed stochastic neighbor embedding (t-SNE) implementations. It is very good at preserving the global structure in the final projection.

As noted in Figure 12 below, there is significant connectivity between the gunshot and plastic_pop sounds [16]. In addition, the doorknock_slam shows a significant amount of connectivity to the gunshot sound [23,24,29,35,41].

Note that the distances between the clusters have no real meaning, as the scale is arbitrary and induced by the optimization approach. The main interpretation that we can deduce from Figure 12 below is that the plastic_pop sound is much more similar to the gunshot sound than it is to the thunderstorm sound, as it is much farther away. From a global perspective, clusters that are closer together are more similar than those that are farther apart.

## 5. Classification

For the classification of gunshot and gunshot-like sounds, the random forest (RF) classifier was chosen because of its easy setup process and its effectiveness as compared to other methods. We will show later in this section that one obtains impressive classification results by only using a small subset of features selected with the MDI and SHAP procedures.

### 5.1. Model for Sound Analysis

RFs are a combination of tree predictors such that each tree depends on the values of a random vector sampled independently and with the same distribution for all trees in the forest [42]. RF consists of a large number of individual decision trees. The trees operate as an ensemble—a method whereby only a concrete, finite set of alternative models is used to obtain a better predictive performance than that from any constituent learning algorithm alone. Each tree in the RF outputs a class prediction, and the class with the most votes becomes the model’s prediction. The low correlation between models (trees) is the key to the success of the RF classifier.

When the RF model is generated, a GridSearchCV is carried out via the scikit-learn library [43]. That is, the hyperparameters of the RF are tuned via an exhaustive search over specified parameter values for the estimator. The parameters of the estimator used to apply these methods are optimized by a cross-validated grid search over a parameter grid. For the train dataset in this research, the approximate time taken for a solution to converge was around 10 min for 2000 samples using a parameter_space = ’n_estimators’: [10,50,100], ’criterion’: [’gini’, ’entropy’], ’max_depth’: np.linspace(10,50,11). Here, ’n_estimators’ is the number of trees in the forest, ’criterion’ is the function to measure the quality of a split, and ’max_depth’ is the maximum depth of the tree. After running the GridSearchCV, the best parameters converged to: ’n_estimators’ = 100, ’criterion’ = ’gini’, and ’max_depth’ = 50.

### 5.2. Data Leakage Avoidance

To help avoid the serious and widespread problem of machine learning data leakage (a process whereby the effects of the train data are transferred to the test data), the train and test data were split prior to any scaling and post-processing steps. For example, the StandardScaler() [43] (a scaling technique that normalizes the features individually or column-wise, to a mean = 0 and standard deviation = 1) function was first fit to the train data and then to the test data prior to the generation of the RF model.

Note that it is not necessary to normalize the dataset, as any algorithm based on recursive partitioning, such as decision trees and regression trees, is invariant to monotonic transformations of the features. In RF, the trees see only ranks in the features based on a collection of partition rules. As a result, there should be no change with scaling. Experiments were conducted using both the raw data and normalized data. As expected, no difference in accuracy was discovered in this analysis. Throughout the evaluations discussed here, the X_train and X_test data were scaled independently.

### 5.3. Feature Reduction

For this configuration, adding the mean and standard deviation to the MFCC, delta, and delta-deltas, using the RF on the full feature set, we obtain the CM shown in Figure 13a below. Taking only the important features according to the MDI feature importance and applying the scikit-learn SelectFromModel, we arrived at 69 features. After the RF classifier was run again with the reduced feature set of 69 features, the CM and classification report (Figure 13b) and the receiver operating characteristic (ROC) with its associated area under the curve (AUC) ((Figure 13c) remained unchanged with a reduction of about 71% in features.

Figure 11 and Figure 14 below show the various figures of merit based on the MDI and SHAP feature importance analysis, respectively. The tabulated list of features used here can be viewed in Table 5 above.

The first 20 features derived from the SHAP feature importance analysis appear to perform marginally better than the MDI feature importance analysis. We also noted from the CM that the gunshot detection is marginally better from the SHAP feature importance analysis as compared to the MDI feature importance analysis.

The list of features from the SHAP analysis that are different from the MDI feature importance analysis are shown highlighted in red in Table 5 above. As noted for the SHAP analysis, most of the features are confined to within about the 10th filter band, whereas the MDI analysis takes us into the 18th filter band. In both cases this shows that the energies used to generate the model need only these low Mel-frequency bands.

## 6. Detection

In this section, we combine the gunshot-like sounds into 1 class. In doing so, we used 200 samples for each of the gunshot-like sounds. This new dataset contains 1800 gunshot-like sounds and 1800 gunshot sounds. Attention was given, as was done above, to ensure that an equal proportion of classes was available for the 80/20 train/test split. In this study, the RF algorithm is again used for its simplicity and effectiveness. Using the GridSearchCV with the same parameters as listed in Section 5.1 above, we ended up with the best parameters converging to: `n_estimators’ = 100, `criterion’ = ’gini’, and `max_depth’ = 34.

Using the important features found in Table 5 above, we generate the confusion matrices for the RF MDI and SHAP analysis (see Figure 15 below).

We also included in Figure 15 above the CM using the full dataset, Figure 15a, and also with only the 16 features that are common to both the feature importance/feature selection analysis, Figure 15b.

We note that Figure 15a,b generate about the same true positive (TP) for the gunshot sounds, whereas the RF MDI and SHAP generate a slightly better TP. This can be accounted for by the fact that the concentrated feature set (the RF MDI and SHAP top 20 features) lends itself to a marginally better model.

Table 6 below compares the accuracy and false positive rate (FPR) (geared toward the gunshot sound) for the various datasets. As can be seen from Table 6, the SHAP top 20 feature set has the best FPR performance and also the best accuracy.

## 7. Conclusions

So far, we have reported the findings of data procurement, analysis, classification, and detection of gunshot and gunshot-like sounds. After an outline of the data procurement procedure, we provided the results of two different feature importance/reduction techniques using MFCC features, i.e., (1) RF MDI and (2) SHAP on gunshot and gunshot-like audio. We also demonstrated the closeness of various gunshot-like sounds to gunshot sounds in the drastically reduced feature space using the UMAP technique. Finally, we presented the classification and detection results using reduced sets of features, in comparison with those obtained with all features.

We showed that the SHAP feature importance process produced marginal improvement over the RF MDI feature importance process. The SHAP analysis generated an accuracy of about 97.78% with the FPR of 0.025 for gunshot detection. The RF MDI produced an accuracy of 97.22% with an FPR of 0.033. Further analyses have discovered that among a total of 240 features, 20 leading features were found to be acceptable to maintain good accuracy and FPR, with a reduction of 92% in terms of the number of features used for gunshot detection. However, both the delta and delta-delta derivatives had to be added to the normal MFCC coefficients to make this result possible. It is interesting to note that among the two sets of 20 leading features identified respectively by the SHAP and MDI algorithms, they shared 16 common features.

From a physical perspective, the most dominant feature in the MDI feature importance was feature 2, which is concentrated around 160 to 359 Hz. The SHAP’s most dominant feature was feature 0, which is concentrated around 0 to 161 Hz. This indicates that much of the information that leads the model to its decision is based in the very low end of the frequency spectrum.

Although not discussed in this paper, an avenue for potentially better detection can be carried out via the pitch shifting of the recorded gunshot events to a higher frequency range. As the MFCC follows the human hearing sensitivity, the higher pitches can become more discernable, and in turn lead to a highly optimized model.

To study how the gunshot-like sounds relate to the gunshot sounds from a global perspective, we employed the UMAP feature reduction technique in the study, which reduced the number of leading features down to two. Viewing the relation to the sounds in this 2-dimensional space gives one a perspective on which sounds are physically audibly close to each other. Further analysis utilizing various methods to isolate these close sounds leads us to conclude that the plastic bag popping sounds resembled the gunshot sounds.

The experimental results of classification and detection further illustrated that employing proper feature importance/reduction techniques can increase the efficiency and improve the overall performance of a gunshot detection model.

## Figures and Tables

**Figure 1 sensors-22-09170-f001:**
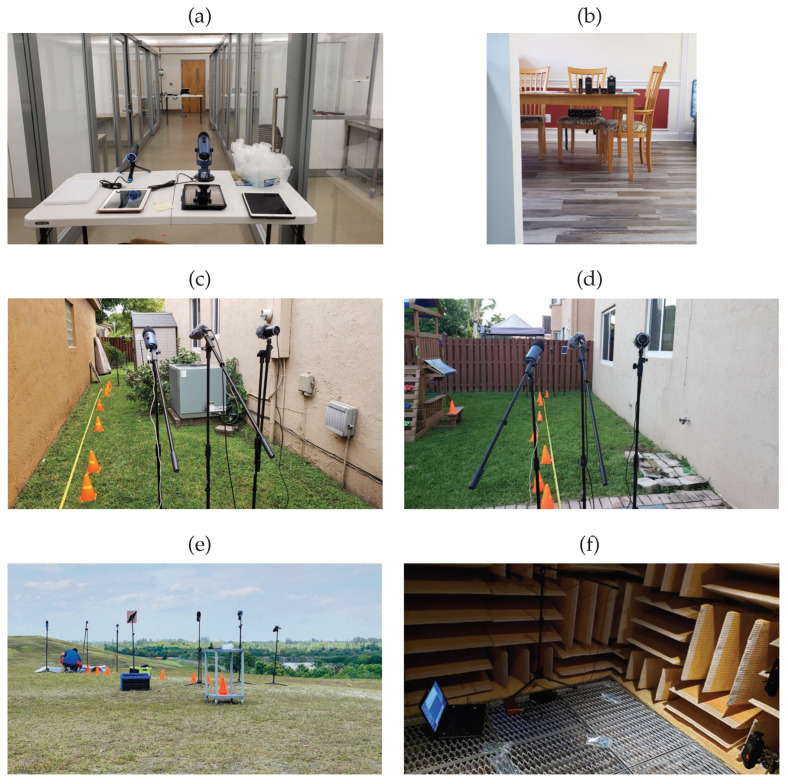
Images of some of the various environments used for the capture of the plastic bag popping sounds (class 8): (**a**) along glass corridor, (**b**) inside of personal dwelling, (**c**) between two buildings, (**d**) side of building, (**e**) open field and, (**f**) anechoic chamber [16].

**Figure 2 sensors-22-09170-f002:**
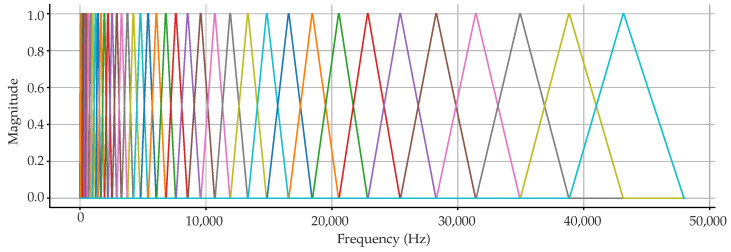
Uniform Mel-bandwidth filterbank for a 96 k sample rate.

**Figure 3 sensors-22-09170-f003:**
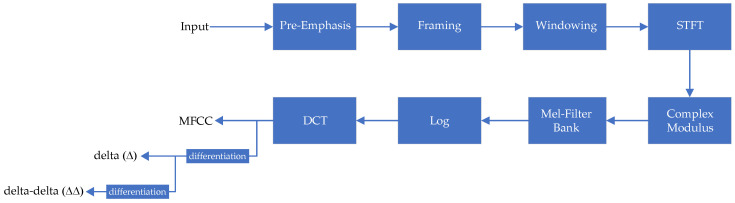
Block diagram of the MFCC process and its derivatives.

**Figure 4 sensors-22-09170-f004:**
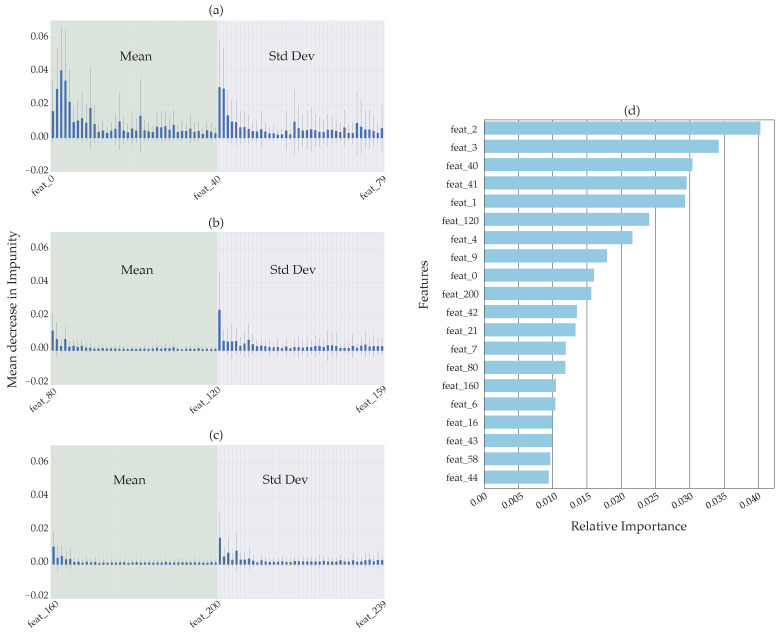
Random forest impurity-based feature importance showing the mean and standard deviation for (**a**) MFCC, (**b**) MFCC-Delta, (**c**) MFCC-Delta-Delta feature extraction and, (**d**) the 20 most important features in descending order.

**Figure 5 sensors-22-09170-f005:**
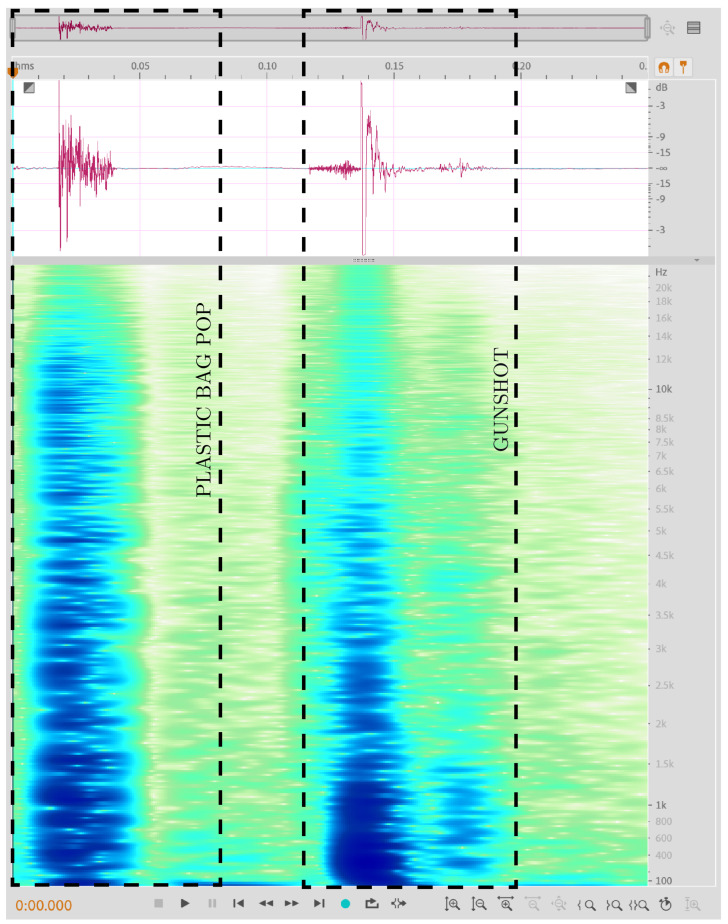
Comparison of zoomed in spectrogram plots of a plastic bag pop and gunshot sound using Adobe Audition.

**Figure 6 sensors-22-09170-f006:**
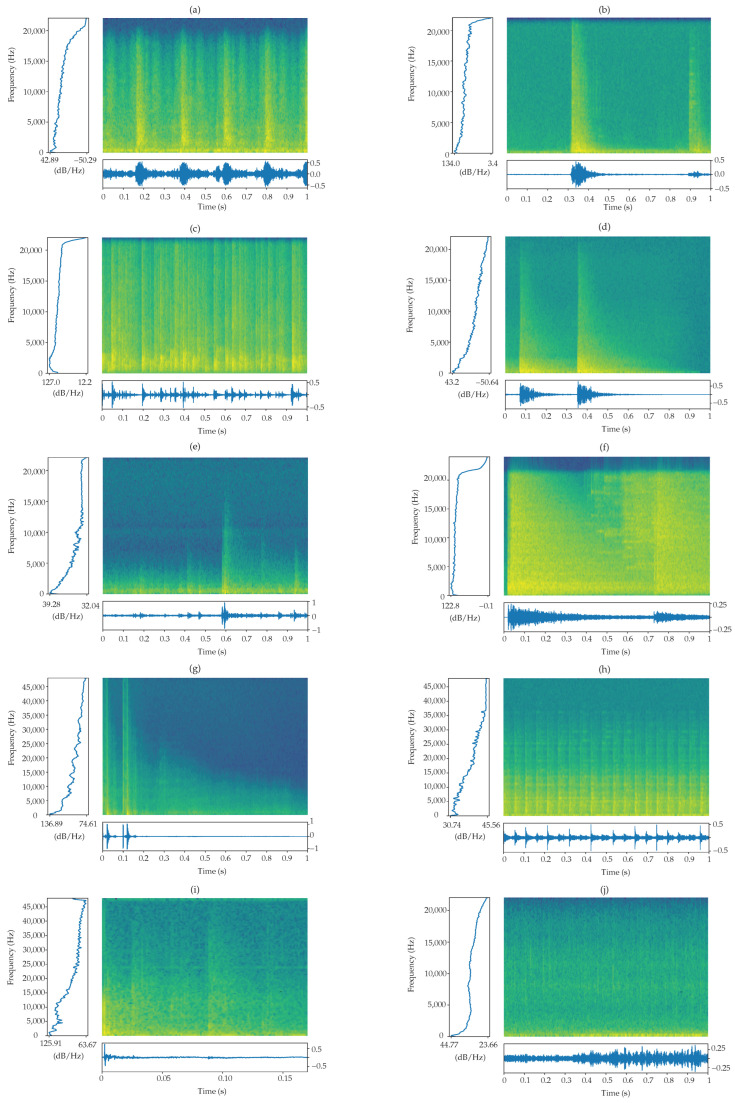
Comparison of the spectrograms and respective PSD for the various classes: (**a**) carbackfire-0, (**b**) cardoorslam-1, (**c**) clapping-2, (**d**) doorknock_slam-3, (**e**) fireworks-4, (**f**) glassbreak_bulb-burst-5, (**g**) gunshot-6, (**h**) jackhammer-7, (**i**) plastic_pop-8 and, (**j**) thunderstorm-9.

**Figure 7 sensors-22-09170-f007:**
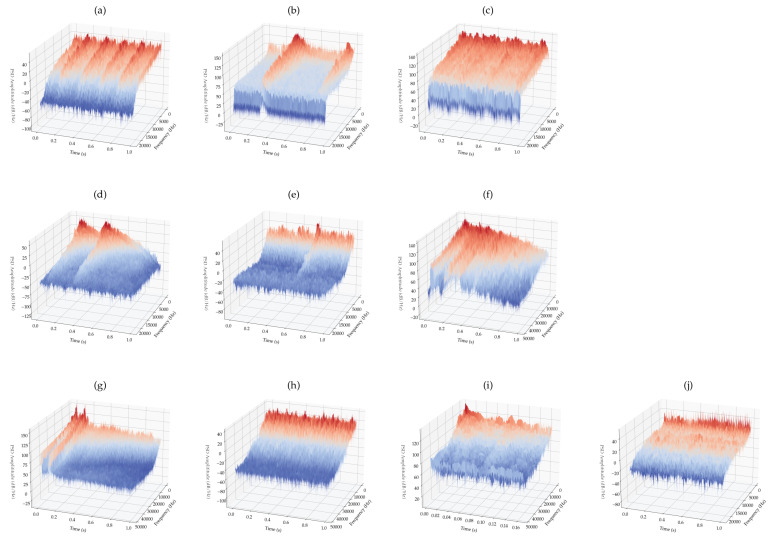
Waterfall plots of the various classes: : (**a**) carbackfire-0, (**b**) cardoorslam-1, (**c**) clapping-2, (**d**) doorknock_slam-3, (**e**) fireworks-4, (**f**) glassbreak_bulb-burst-5, (**g**) gunshot-6, (**h**) jackhammer-7, (**i**) plastic_pop-8 and, (**j**) thunderstorm-9.

**Figure 8 sensors-22-09170-f008:**
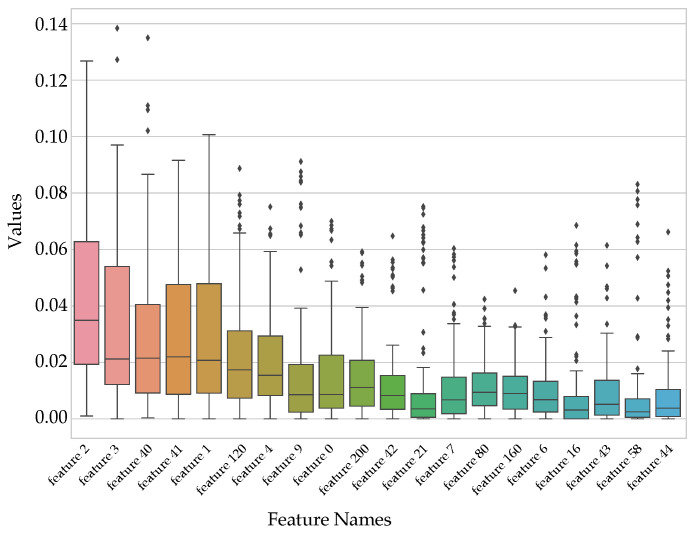
Box plot showing the first 20 sorted MDI features.

**Figure 9 sensors-22-09170-f009:**
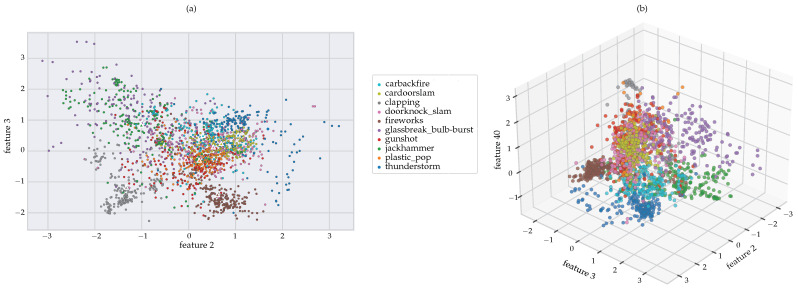
(**a**) 2D scatter plot of feature 2 vs feature 3 with centroids and (**b**) 3D plot of features 2, 3, and 40.

**Figure 10 sensors-22-09170-f010:**
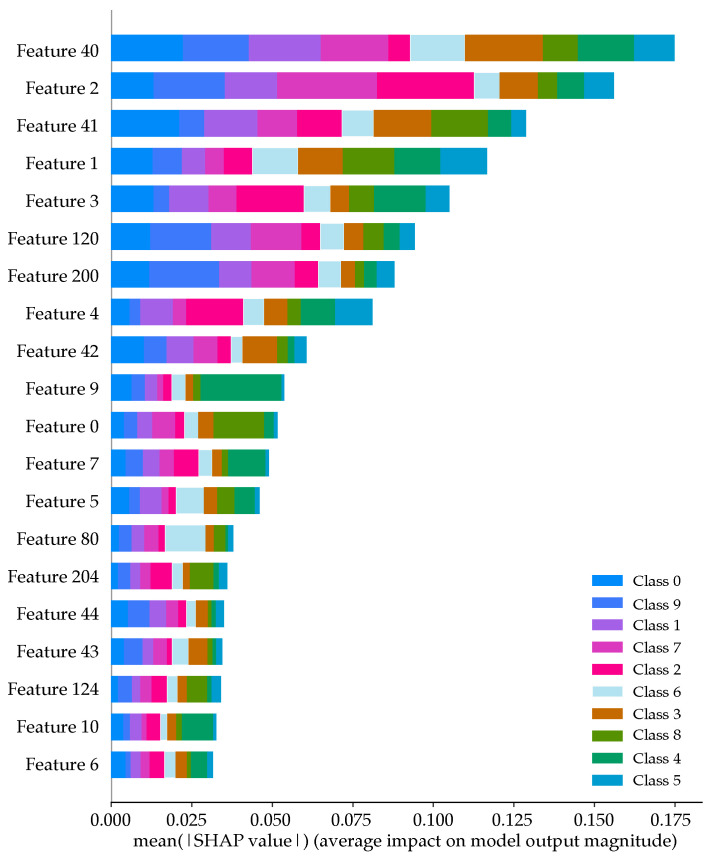
SHAP summary plot.

**Figure 11 sensors-22-09170-f011:**
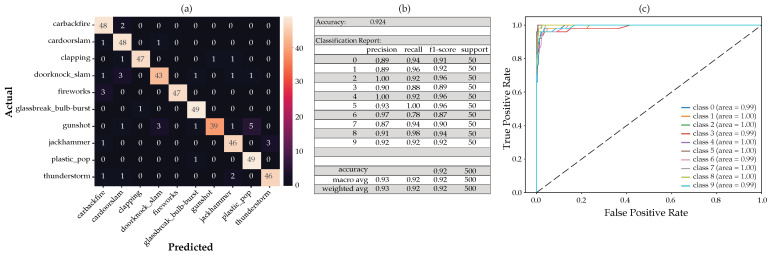
The generated model using only the first 20 features according to the SHAP feature importance analysis, showing the resulting: (**a**) confusion matrix, (**b**) classification report and, (**c**) ROC curves.

**Figure 12 sensors-22-09170-f012:**
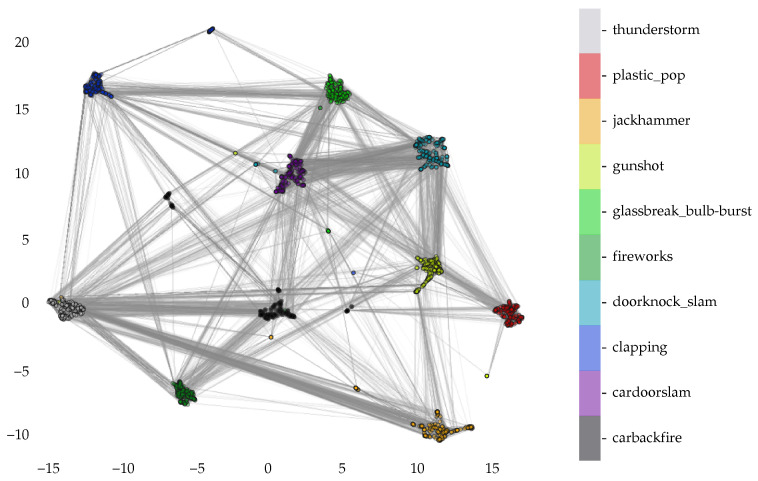
UMAP connectivity for the gunshot and gunshot-like sounds.

**Figure 13 sensors-22-09170-f013:**
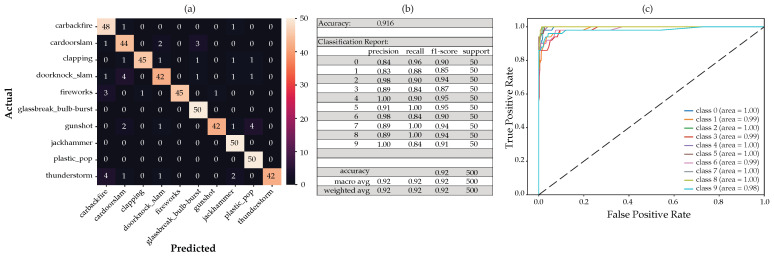
Full feature analysis showing the (**a**) confusion matrix, (**b**) classification report and, (**c**) ROC curves.

**Figure 14 sensors-22-09170-f014:**
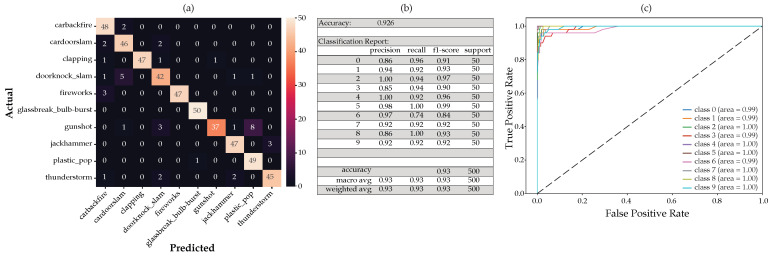
The generated model using only the first 20 features according to the MDI feature importance analysis, showing the resulting: (**a**) confusion matrix, (**b**) classification report and, (**c**) ROC curves.

**Figure 15 sensors-22-09170-f015:**
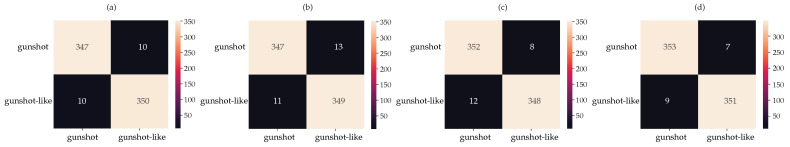
Confusion matrix for (**a**) full dataset, (**b**) 16 common features, (**c**) RF MDI top 20 and, (**d**) SHAP top 20 features.

**Table 1 sensors-22-09170-t001:** Order of the various classes with a partial list of audio used.

carbackfire (0)	cardoorslam (1)
1905 Cadillac	1963 Porsche
1932 Plymouth Roadster car	1964 Cadillac
1967 Doge D100 Adventurer truck	1965 Jaguar E-Type
1971 Ford	1976 Pontiac Grand Prix car
1976 Ford Pinto 3 cylinder car	1983 Chevy pickup truck
1979 Ford F600	1986 Cadillac DeVille sedan
1980 Datsun 210	1988 Chevy pickup truck
Chevy Nova drag racer muscle car	1998 Ford Expedition V8 SUV
	Car trunk
	U-Hall truck
**clapping (2)**	**doorknock_slam (3)**
Applause small group	Wood door hits
Baseball game	Door pounds
Children applause	Elevator door knock
Children in classroom	Glass door
Church applause small group	Glass and wood door
Crowd cheer	Half glass door impact
Crowd clap and stomp	Metal dumpster slam
Crowd rhythmic	Metal screen door slam
Crowd rhythmic fast	Stairwell door slam
Crowd rhythmic scattered	Dublin castle large wood door slam
**fireworks (4)**	**glassbreak_bulb-burst (5)**
New year fireworks in city	Beer bottle break on cement
New year fireworks ambience	Beer smash hit on steel plate
Long intensive	Fluorescent tube crash
Small fireworks	Glass breaking window frame
Mid distance fireworks	Glass picture solid impact
Single burning fireworks bang	Glass safety break
Sparkling single fireworks ambience	Large pickle jar break on cement
Sparkling single fireworks bang	Light bulb smash
	Light bulb smash with hammer
	Plates smash against wall
**gunshot (6)**	**jackhammer (7)**
AK47 bursts	Ambience construction site
Beretta M9	8th floor construction
Glock 9 mm	Urban small construction
M4 double tap	Spread out jackhammer
Maverick 88 single shots	Street construction hydraulic jackhammer
Pistol	Hotel construction, light hammering
Rifle	Short busts
SKS M59 single shots	City industry construction site
Sub machine gun-9 mm	
Winchester 1300	
**plastic_pop (8)**	**thunderstorm (9)**
0.05 m (2in) from Yeti mic using 1.89 L (0.5 Gal) bags-outdoor park	Deep rumble
0.30 m (1FT) from Yeti mic using 1.89 L (0.5 Gal) bags-side of building	Long and slow rolling bursts
0.91 m (3FT) from Tascam mic using 9.08 L (2.4 Gal) bags-between buildings	Long thunderstorm with hard rain
1.52 m (5FT) from JLab mic using 1.89 L (0.5 Gal) bags-inside lab with curtains	Rain and thunder approaching
3.04 m (10FT) from Bruel and Kajer mic using 15.14 L (4 Gal) bags-inside home	Rolling thunderstorm
4.57 m (15FT) from JLab mic using 3.02 L (0.8 Gal) bags-inside lab with glass walls	Storm with strong thunders
6.10 m (20FT) from Zoom mic using 1.89 L (0.5 Gal) bags-inside home	Strong thunderstorm in city
6.71 m (22FT) from JLab mic using 9.08 L (2.4 Gal) bags-inside home	Thunder rumble with constant rain
7.32 m (24FT) from iPad mini using 1.89 L (0.5 Gal) bags-outdoor park	Thunderstorm in closed car
7.32 m (24FT) from Samsung S9 phone using 1.89 L (0.5 Gal) bags - outdoor park	

**Table 2 sensors-22-09170-t002:** Feature numbers and its associated meaning.

Feature Name	Feature Decipher
feature 0….feature 39	MFCC_MEAN_FLTR0…MFCC_MEAN_FLTR39
feature 40…feature 79	MFCC_STDDEV_FLTR0…MFCC_STDDEV_FLTR39
feature 80…feature 119	DELTA_MEAN_FLTR0…DELTA_MEAN_FLTR39
feature 120…feature159	DELTA_STDDEV_FLTR0…DELTA_STDDEV_FLTR39
feature 160…feature 199	DELTA2_MEAN_FLTR0…DELTA2_MEAN_FLTR39
feature 200…feature 239	DELTA2_STDDEV_FLTR0…DELTA2_STDDEV_FLTR39

**Table 3 sensors-22-09170-t003:** Start/Stop frequencies for Mel triangular filters given a sample rate of 96k and 40 Mel coefficients.

Feature	Start (Hz)	Stop (Hz)	Feature	Start (Hz)	Stop (Hz)	Feature	Start (Hz)	Stop (Hz)	Feature	Start (Hz)	Stop (Hz)
0	0	160.97	10	1270.02	1722.96	20	4844.28	6118.98	30	14,903.36	18,490.79
1	76.31	254.80	11	1484.79	1987.10	21	5448.68	6862.35	31	16,604.36	20,582.87
2	160.70	358.88	12	1722.96	2280.03	22	6118.98	7686.76	32	18,490.79	22,903.02
3	254.80	474.32	13	1987.10	2604.90	23	6862.35	8601.04	33	20,582.87	25,476.10
4	358.88	602.33	14	2280.03	2965.18	24	7686.76	9614.99	34	22,903.02	28,329.68
5	474.32	744.31	15	2604.90	3364.74	25	8601.04	10739.48	35	25,476.10	31,494.35
6	602.33	901.76	16	2965.18	3807.86	26	9614.99	11,986.55	36	28,329.68	35,004.01
7	744.31	1076.37	17	3364.74	4299.28	27	10,739.48	13,369.57	37	31,494.35	38,896.27
8	901.76	1270.02	18	3807.86	4844.28	28	11,986.55	14,903.36	38	35,004.01	43,212.85
9	1076.37	1484.79	19	4299.28	5448.68	29	13,369.57	16,604.36	39	38,896.27	48,000.00

**Table 4 sensors-22-09170-t004:** Comparison of relative distances from the gunshot sound to the gunshot-like sounds.

Class	feat2	feat3	Dist from Class 6	Class_Name
6	−0.0525	−0.2625	0.0000	gunshot
8	0.2264	−0.2992	0.2813	plastic_pop
3	0.5276	0.1332	0.7022	doorknock_slam
0	0.3052	0.5041	0.8460	carbackfire
1	0.6965	0.2429	0.9036	cardoorslam
4	0.6124	−1.4075	1.3241	fireworks
9	1.0548	0.5224	1.3573	thunderstorm
7	−1.0090	0.8419	1.4610	jackhammer
5	−0.9919	0.9307	1.5186	glassbreak_bulb-burst
2	−1.3317	−1.1709	1.5689	clapping

**Table 5 sensors-22-09170-t005:** Comparison of the RF MDI and SHAP 20 most important features.

MDI		SHAP	
Feature Name	Feature	Feature	Feature Name
MFCC_MEAN_FLTR2	2	40	MFCC_STDDEV_FLTR0
MFCC_MEAN_FLTR3	3	2	MFCC_MEAN_FLTR2
MFCC_STDDEV_FLTR0	40	41	MFCC_STDDEV_FLTR1
MFCC_STDDEV_FLTR1	41	1	MFCC_MEAN_FLTR1
MFCC_MEAN_FLTR1	1	3	MFCC_MEAN_FLTR3
DELTA_STDDEV_FLTR0	120	120	DELTA_STDDEV_FLTR0
MFCC_MEAN_FLTR4	4	200	DELTA2_STDDEV_FLTR0
MFCC_MEAN_FLTR9	9	4	MFCC_MEAN_FLTR4
MFCC_MEAN_FLTR0	0	42	MFCC_STDDEV_FLTR2
DELTA2_STDDEV_FLTR0	200	9	MFCC_MEAN_FLTR9
MFCC_STDDEV_FLTR2	42	0	MFCC_MEAN_FLTR0
MFCC_MEAN_FLTR21	21	7	MFCC_MEAN_FLTR7
MFCC_MEAN_FLTR7	7	5	MFCC_MEAN_FLTR5
DELTA_MEAN_FLTR0	80	80	DELTA_MEAN_FLTR0
DELTA2_MEAN_FLTR0	160	204	DELTA2_STDDEV_FLTR4
MFCC_MEAN_FLTR6	6	44	MFCC_STDDEV_FLTR4
MFCC_MEAN_FLTR16	16	43	MFCC_STDDEV_FLTR3
MFCC_STDDEV_FLTR3	43	124	DELTA_STDDEV_FLTR4
MFCC_STDDEV_FLTR18	58	10	MFCC_MEAN_FLTR10
MFCC_STDDEV_FLTR4	44	6	MFCC_MEAN_FLTR6

**Table 6 sensors-22-09170-t006:** Tabulated data for the accuracy and FPR using the full dataset, 16 common features, RF MDI top 20, and SHAP top 20 features.

Feature Set	Accuracy	FPR
Full Dataset	0.9681	0.028
16 Common	0.9667	0.031
RD MDI Top 20	0.9722	0.033
SHAP Top 20	0.9778	0.025

## Data Availability

The full dataset used can be found at https://github.com/rbsingh13/Plastic-Bag-Pop-sounds, accessed on 30 September 2022. Please cite this article if you use this dataset for your research.

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
