# Peer review of "Measurements, Analysis, Classification, and Detection of Gunshot and Gunshot-like Sounds"

_sensors, 2022, doi:10.3390/s22239170_

Round 1

Reviewer 1 Report

Your paper is really interesting and you have done a good research work.

Here are my general comments to your paper:

- Indicate better the utility of this research. To try to differentiate real gunshot sounds from other gunshot-like sounds.

- The text in many of the figures (x and y axes in all Figure 6, x axis in Figure 7, legend in Figure 8b, legends and axes in Figures 11, 12 and 13) cannot be read because is too small. You must increase the corresponding font size.

- You should review the format of the references. Some of them are incomplete, the year of publication is missing.

And here are more specific comments:

- Line 99: You should indicate that the anechoic Figure is 1f, not all images in Figure 1.

- Line 101: You have implemented microphones or you have used them?

- Lines 129-130: You should delete the brackets associated to the Sample rate/2 expression.

- Lines 213-215: I don’t understand what you want to express in this sentence. It’s not clear for me. Could you rewrite it trying to clarify the idea that you want to express?

- Line 225: “no features beyond feature 21 are present” Where? This sentence is not clear.

- Lines 228-231: Explanations related with Figure 7 are not clear. Looking at the figure I cannot see the conclusions that you have drawn. Could you explain it better?

- Line 231: “.. guide as to how…” sounds weird to me. I think that it is better “… guide of how…”

- Line 232: “…uncertainty. More…”

- Lines 233-234: I don’t understand clearly the idea that you want to express with this sentence. Could you rewrite or clarify it?

- Figure 7: Why you show the 30 first features, if you have said in the paragraph above that you are going to use only the first 20?

- Figure 8b: z-axis title should show a down to up direction.

- Line 237: It seems that you only show these features in Figure 8, and this figure shows all of them, doesn’t it?

- Line 242: “In Table 4, it is shown/it can be observed…”

- Lines 243-245: What has been verified in these researches exactly?

- Table 4: Feature 2 and 3 columns are the position coordinates of the centroids? It is not clear.

- Line 297: What is the meaning of “t-SNE”? You should include it.

- Lines 329-331: Which are the best parameters that has been selected?

- Line 348: You should indicate that the confusion matrix is (CM), because you use later (line 352) this acronym.

- Table 6: Data shown in this table is already shown in Table 5.

- Line 372: You say that the RF algorithm is used again. Have you used the same model (with the same parameters) than in the previous classification study? If yes, this is correct? Because it was trained to classify 9 classes, and now you have only 2. If not, have you used again cross validation to obtain the best parameters for the learning model? What parameters have you used? You must explain all this information.

- Figure 14: Why don’t you try MDI or SHAP with the 16 common features? And/or, CM with the best 24 parameters (20+4 different). And/or CM with the 20 best features of MDI and CM with the 20 best features of SHAP?

- Line 384: SHAP case shows the best FPR performance and also the best accuracy.

- Line 408: “hones in on”??? I don’t understand that. Verity that this sentence is well written.

Yours sincerely.

Reviewer 2 Report

Why the sampling frequency of recorded events is 96kHz, I think that 48 kHz is enough for this type of events where dominant energy is in low-mid frequency range.

In spectrogram the resolution should be explained better at lower frequencies..?

How is the scaling of amplitude done because all datasets (except in experiments done by authors are not recorded by the same equipment with same sensitivity and the distance from the recorded event is not known.

which type of measurement equipment is used?, sound level meter, microphone, sensitivity in frequency range of interest.

what is the physical explanation for the most dominant feature for recognizing gunshot events?

Have you considered the pitch shifting of recorded gunshot events waveforms in higher frequency range where the human hearing system is more sensitive to get better detectability (regarding MFCC coefficients).
